# Braided Fabrication of a Fiber Bragg Grating Sensor

**DOI:** 10.3390/s20185246

**Published:** 2020-09-14

**Authors:** Songbi Lee, Joohyeon Lee

**Affiliations:** Department of Cognitive Science, Yonsei University, 50 Yonsei-ro, Seodaemun-gu, Seoul 03722, Korea; 9taniasongbi@yonsei.ac.kr

**Keywords:** Bragg grating sensor, auxetic sensor, silica helical core, wrap angle, braid angle

## Abstract

Our objective was to construct textile braiding manufacturing methods to facilitate high precision and accurate measurements using optical fiber Bragg grating sensors for various structures. We aimed to combine three-dimensional (3D) braiding processing with the optical Bragg grating sensor’s accurate metrology. Outside the limits of the sensor’s epoxy attachment methods, the textile braiding method can diversify the scope of application. The braiding process can be used to design a 3D fabric module process for multiple objective mechanical fiber arrangements and material characteristics. Optical stress–strain response conditions were explored through the optimization of design elements between the Bragg grating sensor and the braiding. Here, Bragg grating sensors were located 75% away from the fiber center. The sensor core structure was helical with a 1.54 cm pitch, and a polyurethane synthetic yarn was braided together with the sensor using a weaving machine. From the prototype results, a negative Poisson’s ratio resulted in a curled braided Bragg grating sensor. The number of polyurethane strands was studied to determine the role of wrap angle in the braiding. The 12-strands condition showed an increase in double stress–strain response rate at a Poisson’s ratio of 1.3%, and the 16-strands condition was found to have noise affecting the sensor at a Poisson’s ratio of 1.5%. The findings suggested the application of braiding fabrication to the Bragg grating sensor could help to develop a new monitoring sensor.

## 1. Introduction

Optical sensors have a high-efficiency grating fabrication system that can be used for high-quality inscribing on a single-mode fiber. Helical gratings can open new avenues for applications of orbital angular momentum in the nanoworld. Helical shapes often result from the competition between bending and stretching energies as well as due to a variety of driving forces [1,2,3].

In the femtosecond laser region, precise length measurement experiments were performed based on the principle of optical interference in previous studies. The precision stage can be used, and exact control is possible at 10⁻⁶ μ. However, further research is needed to easily apply the femtosecond laser’s precision Bragg grating sensor [4,5,6,7,8].

The braid structure is an excellent method for studying bending strength, impact resistance, torsion efficiency, and energy absorption. Prior studies reported that composites can be used in the engineering, aerospace, transportation, and medical industries, and various braiding structures and materials have been developed [9].

Braiding new units through the combination of each module allows for the expansion and contraction of complex structures. Multiaxis 3D braided fabrics have multiple layers and the in-plane properties of bias yarn layers [10,11]. 3D braiding makes it possible to weave in various shapes and this is achieved using a woven thread; various weaving-type modules are possible [12,13].

A typical fiber Bragg grating (fbg) sensor is measured by bonding the object to be measured with tensile stress–strain on a hard and flat surface with epoxy. This has limitations in applications when using complex structures or flexible surfaces were using epoxy is impossible. The braiding method is intended to improve the limitations of the practical sensor application. The braiding machine produces helical braiding of the yarn with the combination of the spin group. The helical core is capable of inducing mechanical characteristics that can act as a stress–strain sensor when it is arranged with a helical moving trace of yarn [14,15].

With the fbg sensor, it is possible to derive various application designs for the response axis of the strain. The textualized fbg sensor enables the application of complex structures, and the possibility of the textualized modular sensor enables free combination design applications.

By combining the accuracy of the optical Bragg grating sensor with 3D fabric modular processing in a braiding method [16,17] complicated structures can be measured, and the sensor application can be diversified by presenting the possibility of applying the fabrication to the monitoring measurement method. In the past, hard epoxy molding or epoxy adhesion have been applied for the application of the linear tensile stress–strain response fbg sensor. However, the textile-type sensor may enable wearable applications rather than an attachment.

The objective of this study was to develop woven braiding manufacturing methods to facilitate high precision and accurate measurements using optical fiber Bragg grating sensors in various structures.

### 1.1. Measurement Method of Bragg Grating Sensor

The interrogator is measured by the amplitude and phase value of the Bragg wavelength’s scattered light. According to the photo elastic waveguide response, this deformation is applied to the core by an external physical strain [18]. At this time, the wavelength variation induced by the change in the spacing of the optical fiber Bragg grating, which is measured by the complex factors of the displacement, curvature along the axis, and the elastic point, can be explained by the laws of physics and the imposed load. A detection method capable of quantitative analysis is characterized by external forces changing the center frequency.

When light passes through one period of the Bragg grating, Equation (1) shows the Bragg grid conditions. In Equation (1), neff is the effective refractive index of the optical fiber grating, which means the average refractive index, and Λ is the grating period engraved on the optical fiber.
(1)λBragg = 2 × neff × Λ

When an external physical quantity, such as a short-distance strain of the optical fiber Bragg grating, is applied, the Bragg wavelength is changed by these values, as shown in Equation (2). The Bragg wavelength is determined by the value of the microstructure period and the refractive index (neff) of the core Equation (3) [4]. In bending, the stress is determined from the configuration of the fiber. The strain sensitivity of the Bragg grating can be determined using the Bragg wavelength change [19,20].
(2)$$\left(1-Pe\right)\epsilon \;=\;\frac{\Delta \lambda }{\lambda }$$
(3)$$\frac{\Delta \lambda B}{\lambda B}\;=\;\frac{\Delta \left(neff\;\Lambda \right)}{neff\;\Lambda }\;=\;\left(\begin{array}{cc} 1+\frac{1}{n_{et}} & \frac{\partial neff\;}{\partial \varepsilon } \end{array}\right)\Delta \varepsilon \;=\;ße\;\Delta \varepsilon$$
where ß*e* is the strain sensitivity of the Bragg grating, *Pe* is the photo elastic constant (variation of index of refraction with axial tension), where *Pe* is 0.212.

According to the waveguide, a photo elastic response by deformation is applied to the core by an external physical strain. A wavelength shift induced by the change in the spacing of the optical fiber Bragg grating is measured by the complex factors of the displacement, the curvature along the axis, and the elastic point that can be explained by the laws of physics and the imposed load. It calculates all distributions of reflected light between the start point and endpoint of the sensor peak over the threshold light amount to find the distribution center. The sensor peaks also include specifications for the calculation method suitable for the change in the reflected light amount distribution. It is measured by calculating ∆N and the amount of change in the fractional part ∆*ε* [18,19,20,21].

### 1.2. Braiding Mechanical Structure

During braiding, a bundle of three or more fibers is connected continuously without cutting. All threads are gathered upward in the center of the track and then stretched in the vertical direction to form a braid. There are two or more tracks, each with a group of spindles moving in different directions. The angled braid is characterized by continuous threads with a structured design element that becomes a columnar structure. 3D weaving of complicated purpose fabrics is possible and has been applied to various industries [22]. Braiding, one of the methods used for weaving composite fibers, can be fabricated in the desired direction depending on the strength and stiffness of the desired structure. The braid enables the continuous orientation of the fibers, allowing the mechanical properties’ design according to the properties of the material and construction [23]. The braiding angles of the yarns in different directions are not identical. The hybrid braiding angles introduce geometrical incompatibility into the longitudinal deformation of the structure, which leads to improved longitudinal stiffness without compromising the bending. The constant and varied braid angles on the conical mandrel’s surface determine the strength of the twist, and the mechanical properties can be designed using an auxiliary structure [24]. 

#### Negative Poisson’s Ratio

Poisson’s ratio is a characteristic of a material. When a force acts on a material, the material’s deformation occurs in the direction in which it is applied. If the tensile force acts on the material, it is stretched in the direction of the tensile force. It is presented in Figure 1. The Poisson’s ratio represents the ratio between the horizontal strain and the vertical strain Equation (4). In other words, when the tension decreases in the direction that the load moves and increases in the vertical direction, the material has a positive Poisson’s ratio. In Figure 1 and Figure 2, the principle of Poisson’s ratio was explained by connecting the angle and pitch of the wrap angle with tubular braiding. In this experiment, two braided angles and wrap angles were applied to explore the structural principle that negatively affects Poisson’s ratio.

A material tends to thin when stretched. Rarely, some materials contract or stretch when stretched horizontally, indicating a negative Poisson’s ratio (PR). PR is called auxetics. A negative Poisson’s ratio enables the stiffness and resistance to be actively increased to respond to external stress action loads [25,26,27,28].
(4)$$\nu \;=\;\left|\frac{Lateral\;strain}{Axial\;strain}\right|\;=\;-\frac{\varepsilon_{y}}{\varepsilon_{x}}\;=\;-\frac{\varepsilon_{z}}{\varepsilon_{x}}$$

## 2. Materials and Methods 

### 2.1. Design of the Braided Sensor

The core of the sensor used for this study was designed to optimize the blade. This experimental fiber core was located outside of the fiber at half the fiber volume (35 µm from the center of the fiber). It was a helical core with a diameter of 6.3 µm, had a numerical aperture of 0.21, and a core validity index within 0.05%. It was rotated at high speed in the preforming step and twisted during the process to have a pitch of 15.4 cm at 50 times per meter and was coated with a transparent acrylate-based UV fiber. The core applied to the sensor was twisted positively at a twist rate of 0.49π to form an optical waveguide in the clockwise direction. The fiber was treated with germanium doping, producing a Bragg grating sensor through precision processing using a femtosecond laser process through a phase mask. It had an operating wavelength of 1550 nm, a coating diameter of 185.7 µm, and an optical fiber diameter of 125.6 µm.

The Bragg grating engraved sensor was attached to the plastic optical fiber with epoxy. When weaving tubular blading, it was fabricated by inserting it into the bladder core yarn with irradiation of elastic pieces. The outer yarn was made of polypropylene fiber, and the blade angle was woven with a uniform distribution of 45°. The elastic knitting was set so that the number of strands for the same yarn (12 and 16 strands) could be uniformly distributed to the braided core yarn. The irradiated elastic piece was spirally wound around the tubular bladder core yarn, and an initial angle value of 45° for blading was used at four uniformly distributed points. All threads were assembled upwards in the center of the orbit and bladed vertically. Three orbits were used, with a blade machine with eight spindle groups moving in different directions.

Figure 2 shows the prototype used in this experiment. The braiding was composed of polyester nylon. Polyurethane yarn was used in between the braiding yarn and the flexible sensor rod-body frame to maintain the uniform internal tension.

### 2.2. Methodology

A comparative analysis was conducted on the sensor’s signal responsivity with the epoxy attached at the flexible rod-body frame sensor and with a braiding sensor. The braiding conditions were analyzed by the size of the sensor signal and the morphology changes’ reliability. The valid responses of the sensor in the context of the conditions of the braiding and wrap investigations were analyzed based on the changes in internal shear stress to calculate the range of the investigated braid state. 

### 2.3. Experiment

The prototype was subjected to a two-step manufacturing process: the ultra-precise femtosecond laser process was the primary process. The produced Bragg grating sensor was combined with a polyurethane yarn in a braiding machine and then went through a secondary braiding process.

#### 2.3.1. Prototype 

##### Bragg Grating Sensor Process 

Figure 3 shows the femtosecond Bragg grating process. A helical core structure optical fiber (Fiber Core™, Southampton, UK) was used for the same braiding yarn angle array. A femtosecond laser was used for precision Bragg grating processing. The femto laser processing specifications were: femtosecond laser grating-pulse repetition rate of 1 kHz, output 2 W (output power), time width 30 fs–2 ps variable conditions, center wavelength of 785 nm, pulse width of 185 fs, and femto laser with maximum output of 1 W (model-APRI Femto-k1, Advanced Photonics Research Institute, GIST, South Korea). These were used to perform the Bragg grating processing of the phase mask treatment. In the Bragg grating processing, scan-speed was 2 mm/s, the line spacing was 2µ, the processing depth (Z) was fixed at −60 µm, and the laser power was changed from 0.3 to 3 mW. While processing the Bragg grating, the signal at 10 kHz of the interrogator was detected in real-time, and the peak generation and shift of the wavelength were observed.

##### Braiding Process 

After attaching the epoxy to the fabricated sensor’s flexible body frame (cross-section diameter 1.2 mm, length 100 cm), arranged so that polyurethane uniformly covers the flexible load, the braiding machine (model-KV-BM12, Zhengzhou Kovi Machinery Co., Ltd., Henan, China) was bonded together with the core yarn at the core yarn position. The 12- and 16-strand polyurethane thread prototypes were made. Figure 4 is a schematic diagram of the braiding machine used to make the prototypes and shows the principles of the braiding method.

#### 2.3.2. Prototypes A, B, C

Three prototypes were produced: flexible load sensor before the braid (Prototype A), the 1- strand braided sensor (Prototype B), the 16-strand braided sensor (Prototype C). A detailed description of the prototype is presented in Table 1. As a result of prototyping, the 12-strand braided sensor induced cyclic curling with a diameter of 6 cm, and the 16-strand braided sensor was produced with cyclic curling with a diameter of 3 cm.

#### 2.3.3. Subject

The pre-braiding sensor (flexible rod-body; prototype A), 12-strand polyurethane braiding (12 strands; prototype B), and a prototype with 16 strands of polyurethane braiding (16 strands; prototype C) were compared to investigate the effect of braiding on the sensor and to explore the conditions for braiding. The sensor unit length (1 cm) and angle (1°) were measured before and after the blading. The maximum and minimum values of the stress–strain response was recorded.

#### 2.3.4. Experimental Device and Tools

The sensor strain measurement behavior was controlled using a device in Figure 5. The Angle and length adjustment range for the stage setup was used. The device can be adjusted manually and has a 1 mm and 1° adjustment resolution. The metronome controlled the speed of the experiment. The 10 kHz measurable interrogator was used to record real-time measurements. 

#### 2.3.5. Protocol

The stress–strain response was investigated through repeated experiments with increasing and decreasing the unit length and angle. In addition, the maximum and minimum values of the stress–strain response were investigated. The reliability coefficient was used to verify sensor reliability (internal consistency) in the repeated experiment. The experimental limiting condition involved using an axial fixer at both ends of the prototype to ensure that the optical fiber axis was fixed. The length of the moving line and the optical waveguide axis move on one plane. As shown in Figure 6, three sets of experiments were performed at metronome 60 at a regular interval of 1 cm and at 10°. According to the repetitive motion, the reliability of the test is determined using the reliability coefficient.

## 3. Results

Braiding allows the continuous orientation of the fibers so that the mechanical properties can be designed according to the properties of the material and structure. The yarn textile packaging design of Bragg grating sensors is more sophisticated than conventional electronic sensor measurement methods and has advantages in terms of noise stability against the external environment. The impact of packaging is just as crucial as its delicate measuring ability. A study on the Poisson ratio condition of the range of Bragg grating sensors, which allows the validity of the signal value to be observed, was performed based on Braiding’s polyurethane yarn count condition. It was investigated by measuring the induced internal tension and strain variation of the signal.

The experimental results showed a positive braiding range and interconnected relationship with the stress–strain response to the fast waveguide axis strain. These results were used to propose the application standards for sensor braiding. Strand cords were constructed to have helical curling using rubber elasticity. In Figure 5A–C, with the same braid conditions, the number of polyurethane yarns caused a change in the internal stress and a negative Poisson’s ratio. As a result, the 16-strand sensor was made with a curling diameter of 3 cm, and the 12-strand sensor was made with a curling diameter of 6.6 cm. The 16-strand sensor had pre-strain, and we observed noise in the morphology. The 16-strand sensor was investigated for non-conformance conditions, and the results are presented in Figure 5.

The 12-strand sensor was observed to have maximum and minimum wavelength shifts with the same morphology as the flexible rod-body frame sensor condition, so the 12-strand sensor testing was used with a flexible rod-body frame sensor. The result was analyzed using wavelength shift rates before and after braiding. Post-braiding, the loading body frame sensory strain response rate decreased by 0.4 times in the 12-strand sensor, while the angle was reduced by 0.3 times in Figure 7. The maximum and minimum values were the same before and after braiding, but the sensor reactivity (wavelength shift speed) increased two-fold in Figure 7.the data of the experiment are summarized in Table 2, and the statistical results are presented in Appendix A.

The decrease in the stress–strain response of the angle and length displacement can be interpreted as increased pressure for the sensor motion. However, in the internal polyurethane 12 strands condition, the sensor reactivity increased two-fold due to internal helical auxetic stress in Figure 8, Figure 9 and Figure 10.

Increased strain reactivity was due to the wrap yarn role of polyurethane, which resulted in a positive interaction and a fast stress–strain response from the helical auxetic stress in the Bragg sensor core. This is presented in Figure 9 and Figure 10. The range of the helical auxetic stress was calculated using the conditions for the 12- and 16-strand sensors. The impact range of the helical auxetic distortion rate in the sensors was analyzed in connection with the fiber optic sensor’s cores pitch cycles (Figure 11 and Figure 12).

The fiber optic cores have a cycle of 1.54 cm with a period of 0.49π. From 0.5π to 0.6π, 6 cm torsion caused the internal tension on the Bragg grating in the sensors to increase two-fold; therefore, the reactivity was increased by twice as much compared to before braiding. However, the 16-strand sensor had a period of 0.7π and helical structures with an internal diameter of 3 cm, which exceeds the inner tension in the sensors and results in noise being measured. The 12-strand sensor, with a helical diameter of 6 cm and a sensor optical waveguide’s vertical section of 3 mm, was used. In Figure 11, the 16-strand sensor with a helical diameter of 3 cm and an optical waveguide’s vertical section of 35 mm is used. The state inside the bladed sensor is shown in Figure 12. Each measured value is recorded in Table 2.

## 4. Discussion

In the general linear tension-type Bragg grating, the sensor’s measurement sensitivity may vary depending on whether the sensor and the object are integrated by epoxy or molding. This study’s helical core can be measured in a situation where there is no complete contact with the subject. Directional angle measurement and displacement measurement are possible, suggesting the possibility of vector-based directional displacement measurement for chiral motion in Figure 7 and Figure 8. A study on deriving the possibility of a suitable range of tension for a grating in a Bragg grating sensor can lead to an application study with many limitations.

Through the results mainly covered in the experiment, the range of the helical auxetic stress was calculated using the conditions for the 12- and 16-strand sensors. The impact range of the helical auxetic distortion rate in the sensors was analyzed in connection to the cycles of the fiber optic sensor’s cores pitch. Poisson’s ratio can be used to interpret the tension range impacting the fiber optic cores.

The normal tension in Figure 13 represents the strain placed on the internal Bragg grating based on the same modification rate of 1.1%. The sensor’s inner tension was calculated within a range (1.3%) that would produce a mutually beneficial relationship at the same ratio of increase as the critical point tolerance range. This is demonstrated by the positive auxetic tension of Figure 13. The negative auxetic pressure diagram outlines Bragg grating conditions when they increase to a point outside of the critical limit (1.6%) as a result of internal compression. 

## 5. Conclusions

Using a braid weaving machine, a core loom center with a sensor and polyurethane composite yarn was braided. The braided sensor curled as a result of having a negative Poisson’s ratio. The mutually beneficial relationship range and noise range of the sensor were differentiated based on the number of braided polyurethane strands. For the 12-strand sensor, a Poisson’s ratio of 1.3% resulted in a two-fold stress–strain reactivity increase. For the 16-strand sensor, a 1.5% Poisson’s ratio resulted in pre-strain noise. Poisson’s ratio for the range strain responses was analyzed as braiding tensor factors that impacted the core, and they can be described using three different levels. Level 1 is defined as the external stress–strain factors that occur only until the point of cladding. At this level, there is no tensor identified at the core that results in wavelength shifts in the Bragg grating. Level 2 is defined as the fiber optic stress–strain factors that occur up to the outer levels of the core’s surface. At this level, tensors in the core that result in wavelength shifts were determined to be horizontal factors. Level 3 is defined as the strain that occurs within the internal core to 50% of the center in the same direction as the light wave axis. At this level, tensors in the core result in wavelength shifts that were determined to be both horizontal and vertical factors. In terms of conditions for braiding conditions, the requirements for a mutually beneficial relationship in the sensors were explored. The conditions for mutually beneficial relationships were confirmed to be a Poisson’s ratio of 1.3% at Level 3. 3D fabrication modules due to braiding may present prospects for the application of sensors in complicated and convoluted areas of design.

## Figures and Tables

**Figure 1 sensors-20-05246-f001:**
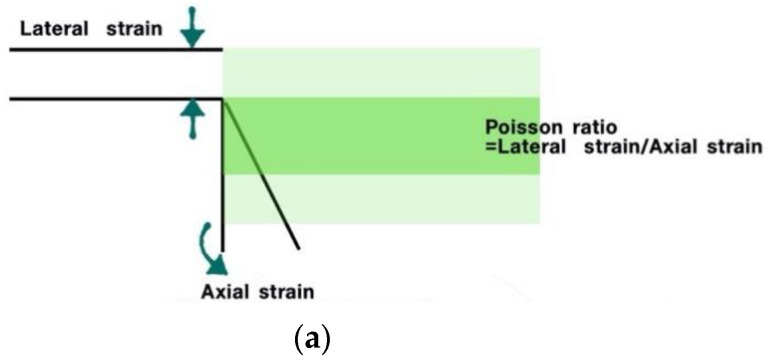
(**a**) A drawing to help understand the definition of Poisson’s ratio. (**b**) The lap angle, lap yarn pitch, and the circumferential length of the braided cross-section, causing a negative Poisson’s ratio in tubular braiding.

**Figure 2 sensors-20-05246-f002:**
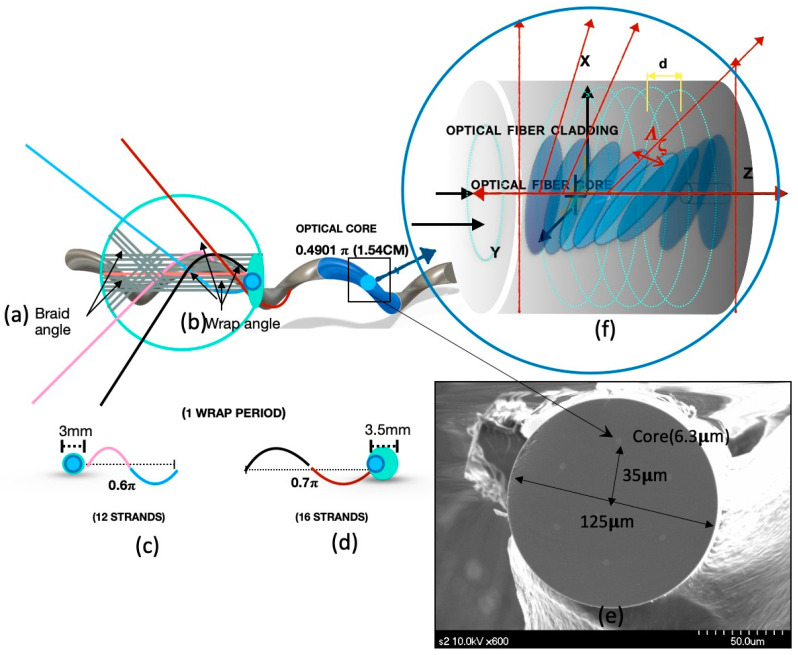
The correlation between the fiber Bragg grating sensor and the braiding angle and wrap angle. An example of (**a**) is the braiding angle during the braiding process and (**b**) the polyurethane wrap angle. (**c**) The wrap pitch of 0.6π and the sensor cross-section diameter of 3 mm for the 12-strand polyurethane sensor. (**d**) The 16-strand polyurethane wrap pitch of 0.7π and the sensor cross-section diameter of 3.5 mm. (**c**,**d**) The samples after brazing, and (**e**) electron micrograph of the braided sensor. It also shows a core position in the optical fiber with a 125 µm diameter and the Bragg grating sensor core with a 6.3 µm diameter. (**f**) The process of the stress–strain response applied to the Bragg grating sensor through the braiding process.

**Figure 3 sensors-20-05246-f003:**
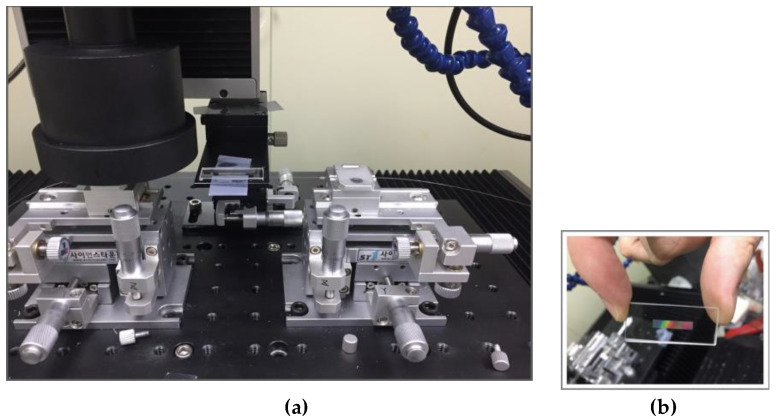
(**a**) A photorealistic view of processing Bragg gratings in a femtosecond laser precision processing stage using a phase mask. (**b**) Photograph of the phase mask (period: 2132 nm, type: uniform, spacing: 2 nm, size: 25 × 3 mm, illumination: 808 nm) used for femtosecond laser processing.

**Figure 4 sensors-20-05246-f004:**
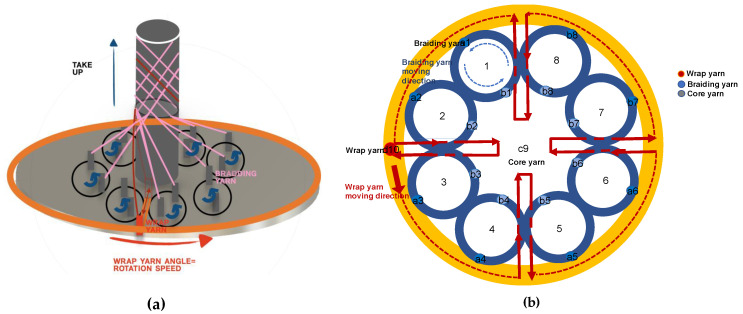
(**a**) The overall structure of the braiding machine. (**b**) The trajectory of the yarn movement of the braiding machine. Eight groups of braiding yarn tracks and one wrap yarn track used for braiding create a combination of spindle movements by repeating the same speed. The core yarn in c9 is surrounded by a tubular braiding structure and by a spindle combination of braiding yarn and wrap yarn. The sensor is inserted along with the core yarn and processed into a tubular bladed process.

**Figure 5 sensors-20-05246-f005:**
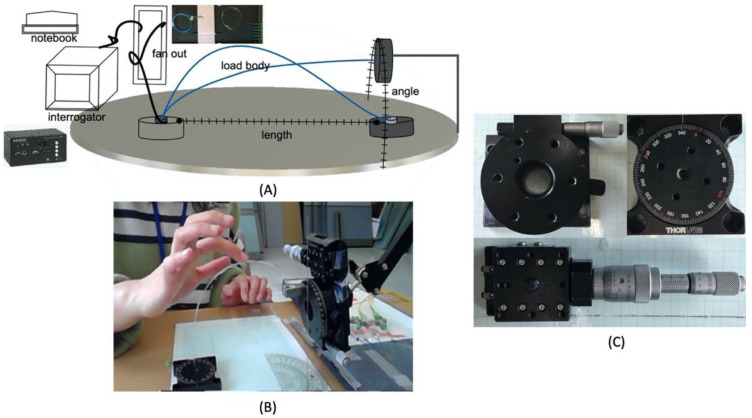
The measurement equipment used in the experiment. (**A**) Structure of experimental measurement. (**B**) Actual photo of experimental measurement. (**C**) Manual measuring tool. The fan-out can accurately connect the laser and Bragg wavelength shift to the interrogator with a Bragg grating core. optical manual device can measure the length (1 mm) and angle (1°) with precision resolution. 10 kHz interrogator ran for real-time monitoring.

**Figure 6 sensors-20-05246-f006:**
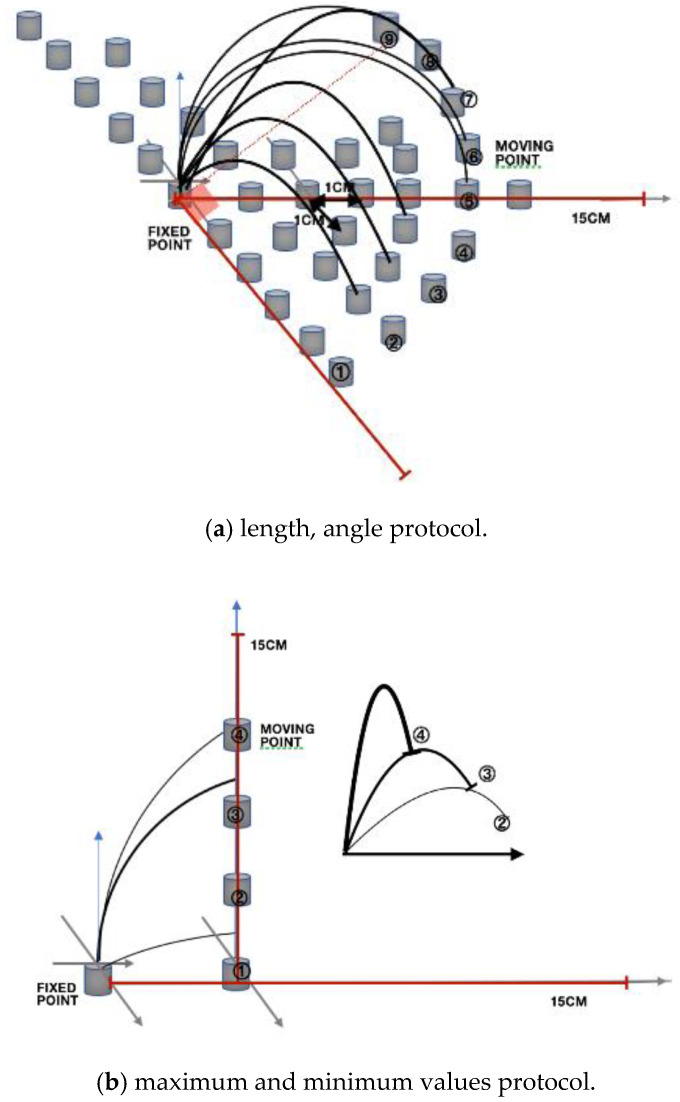
(**a**) Experimental measurement device with a length unit of 1 cm and an angle unit of 10°. (**b**) A diagram of the experimental measurement device for measuring the maximum and minimum values. Each fixed position and moving position were selected, and rubber tabs for the fixed optical axis were installed at both ends of the sensor.

**Figure 7 sensors-20-05246-f007:**
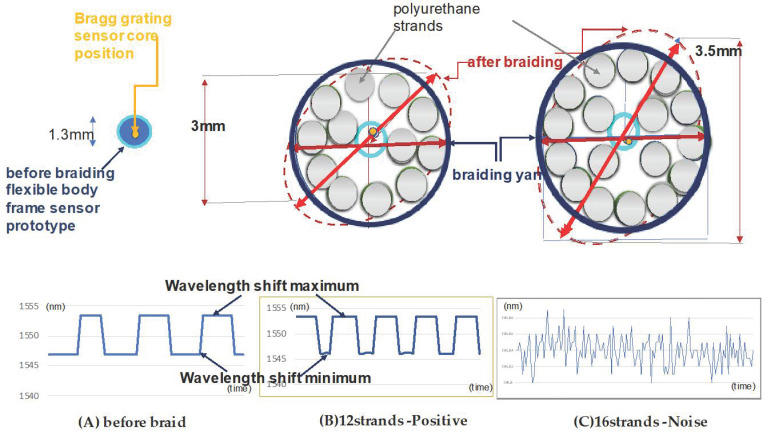
The wavelength variation for the change in the maximum and minimum values was measured; the morphology of the braid. Before the braid (**A**) and after the 12-strand braid (**B**) show the effectiveness of the sensor (**C**). The noise signal in the 16-strands braid.

**Figure 8 sensors-20-05246-f008:**
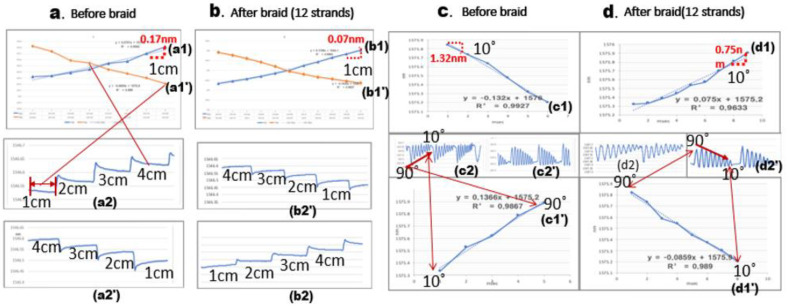
The morphology of the sensor’s length and angle unit wavelength variations and the maximum and minimum values of the sensors before and after braiding under the same conditions (**a1**,**a1’**,**a2**,**a****2’**,**c1**,**c1’**,**c2**,**c2’**). The linearity of the wavelength variation through the trend line. (**b1**,**b****1’**,**b2**,**b2’**,**d1**,**d1’**,**d2**,**d2’**): a1, a1’, b1, b1’: 1cm stress-strain response graph for displacement. a2, a2’, b2, b2’: The morphology of the wavelength shift (**a1**,**a1’**,**b1**,**b1’**) in real time (nm/msec) measured in the interrogator. (**c1**,**c1’**,**d1**,**d1****’**): 1° stress-strain response graph for displacement. (**c2**,**c2’**,**d2**,**d2****’**): The real time (nm/msec) morphology of the wavelength shift (**c1**,**c1’**,**d1**,**d1****’**) in the interrogator.

**Figure 9 sensors-20-05246-f009:**
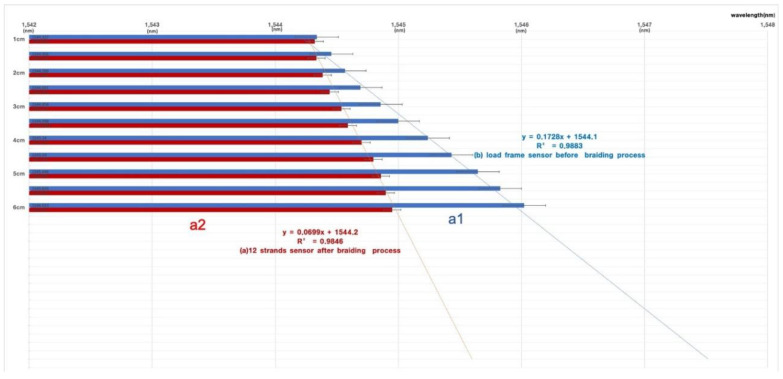
The result of Figure 6. The reliability value of each experiment is presented by the reliability constant for each result value. The stress–strain response change before and after braiding. (**a1**) The wavelength shift for 1 cm displacement before braiding. (**a2**) The wavelength shift for the unit (1 cm) displacement after braiding.

**Figure 10 sensors-20-05246-f010:**
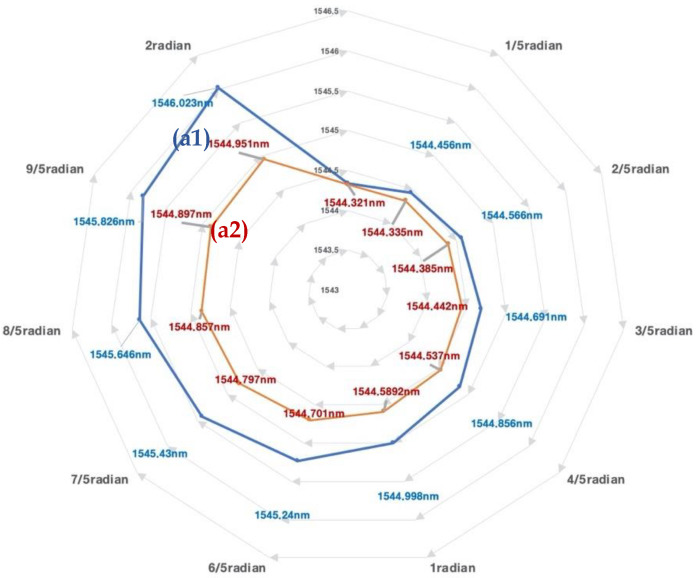
(**a1**) The sensor stress–strain wavelength shift before braiding. (**a2**) The wavelength variation after braiding. The wavelength shifts according to 1 radian.

**Figure 11 sensors-20-05246-f011:**
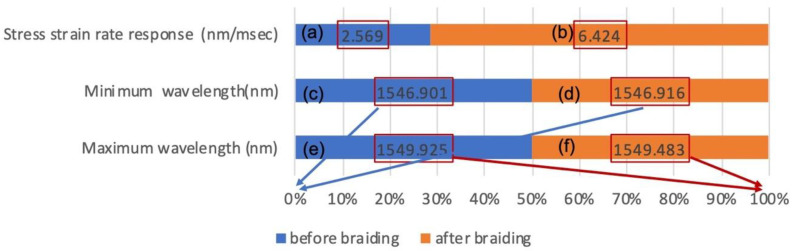
Comparison of the stress–strain response of the optical waveguide axis before and after braiding (**a**,**b**). The fast rate of responsiveness to stress–strain. (**c**,**d**) The stress–strain response minimum (0%) before and after braiding on the optical waveguide axis. Before and after the stress–strain response, maxima (100%) were compared (**e**,**f**).

**Figure 12 sensors-20-05246-f012:**
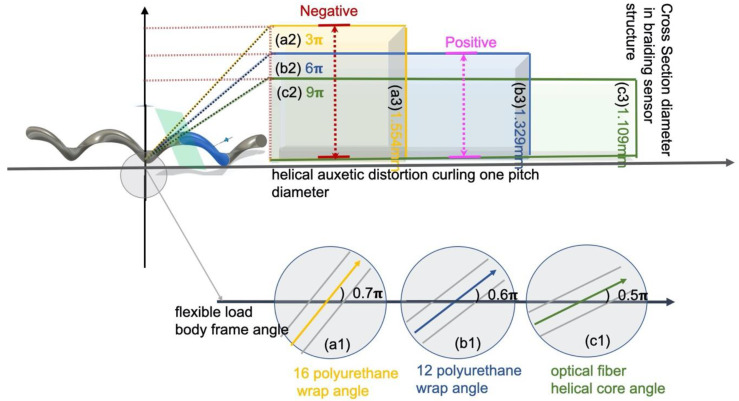
The experimental results analysis of the sensor braiding condition. (**a1**–**a3**) Prototype A condition. (**b1**–**b3**) Prototype B condition. (**c1**–**c3**) Prototype B condition. (**a1**,**b1**,**c1**): Braiding angle relationship between sensor angle and amount of polyurethane; (**a2**,**b2**,**c2**): One pitch diameter curling spiral secondary distortion structure diameter; (**a3**,**b3**,**c3**): Braid sensor diameter Cross section structure diameter.

**Figure 13 sensors-20-05246-f013:**
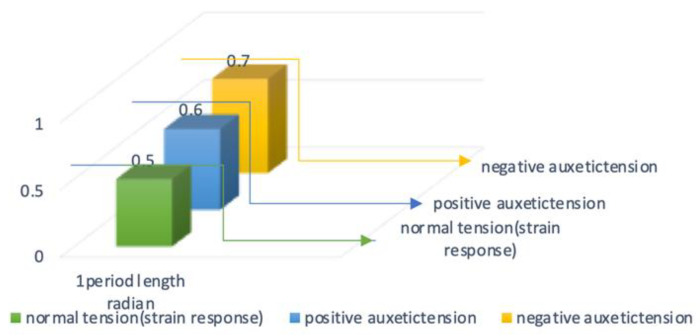
Analysis of sensor stress–strain signal results according to braiding conditions.

**Table 1 sensors-20-05246-t001:** The states of the sensor before and after the braid. Table 1c, d show the cases where 1 strand of polyurethane is braided, and 16 strands of polyurethane are braided. The diameter of the cross-section of the sensor and the diameter of the helical size are indicated.

	Types	Prototype A	Prototype B	Prototype C
Parameter				
Sensor fabrication conditions		flexible rod-body frame	12 strands	16 strands
Cross-section diameter (cm)		0.2	0.3	0.35
Initial braid angle (degree)		No	45	45
Number of polyurethane strands (number)		No	12	16
Braided sensor body curling diameter (cm)		9	6	3
**Prototype photos**				
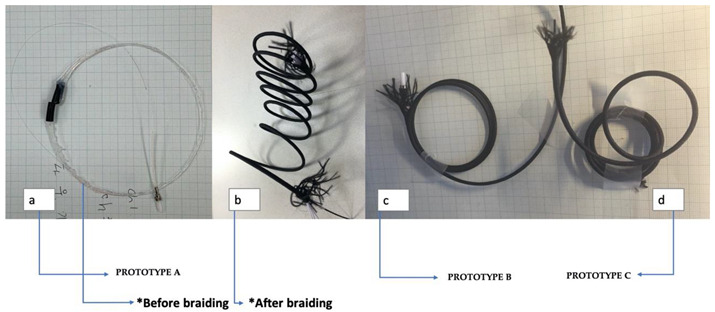				

**Table 2 sensors-20-05246-t002:** The data for the three prototypes investigated (length, angle, maximum and minimum values, confidence constant—Cronbach’s α).

	Subject	Length (Wavelength Shift/1 cm)				
Prototype						
flexible frame		1 cm 6 cm	2 cm 7 cm	3 cm 8 cm	4 cm 9 cm	5 cm 10 cm
		1547.337 nm 1547.563 nm	1547.409 nm 1547.568 nm	1547.44 nm 1547.594 nm	1547.519 nm 1547.619 nm	1547.53 nm 1547.63 nm
		*y* = 0.172*x* + 1575.2 *R*^2^ = 0.9562 (1st, 2nd, 3rd mean)				
		Standardized Cronbach’s α 0.894 (3rd) (α ≧ 0.9 (excellent), 0.8 ≦ α ≦ 0.9 (good), 0.7 ≦ α ≦ 0.8 (acceptable))				
12-strand braid		1575.321 nm 1575.63 nm	1575.392 nm 1575.71 nm	1575.465 nm 1575.79 nm	1575.52 nm 1575.86 nm	1575.355 nm 1575.948 nm
		*y* = 0.0701*x* + 1575.2 *R*^2^ = 0.9562 (1st, 2nd, 3rd mean)				
		Standardized Cronbach’s α 0.861 (3rd) (α ≧ 0.9 (excellent), 0.8 ≦ α ≦ 0.9 (good), 0.7 ≦ α ≦ 0.8 (acceptable))				
16-strand braid		NO				
		angle (wavelength shift/1°)				
flexible frame		0° 50°	10° 60°	20° 70°	30° 80°	40° 90°
		1547.7 nm 1547.561 nm	1547.671 nm 1547.538 nm	1547.611 nm 1547.527 nm	1547.597 nm 1547.496 nm	1547.571 nm 1547.458 nm
		*y* = 0.134*x* + 1547.4 *R*^2^ = 0.9827 (1st, 2nd, 3rd mean)				
		Standardized Cronbach’s α 0.885 (3rd) α ≧ 0.9 (excellent), 0.8 ≦ α ≦ 0.9 (good), 0.7 ≦ α ≦ 0.8 (acceptable), 0.6 ≦ α ≦0.7 (questionable), 0.5 ≦ α ≦ 0.6 (poor), α ≦ 0.5 (unacceptable)				
12-strand braid		1547.453 nm 1547.568 nm	1547.467 nm 1547.594 nm	1547.493 nm 1547.612 nm	1547.512 nm 1547.63 nm	1547.563 nm 1547.451 nm
		*y* = 0.0233*x* + 1547.4 *R*^2^ = 0.9827 (1st, 2nd, 3rd mean)				
		Standardized Cronbach’s alpha. 0.884 (3rd) α ≧ 0.9 (excellent), 0.8 ≦ α ≦ 0.9 (good), 0.7 ≦ α ≦ 0.8 (acceptable), 0.6 ≦ α ≦ 0.7 (questionable), 0.5 ≦ α ≦ 0.6 (poor), α ≦ 0.5 (unacceptable)				
16-strand braid		NO				
		maximum and minimum values of the stress–strain response				
flexible frame		0° 1546.901 nm	90° 1549.925 nm	*y* = −3.567*x* + 1554.1 *R*^2^ = 1 (1st, 2nd, 3rd mean)		
12-strand braid		1546.916 nm	1549.483 nm	*y* = −6.424*x* + 1559.7 *R*^2^ = 1 (1st, 2nd, 3rd mean)		
16-strand braid		NO				

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
