# Peer review of "Braided Fabrication of a Fiber Bragg Grating Sensor"

_sensors, 2020, doi:10.3390/s20185246_

Round 1

Reviewer 1 Report

Specialist in sensor technology used to read journal Sensors. Therefore, well known common statements as    Fused silica…………..lines 27- , are redundant. Please, eliminate also similar sentences (e.g. lines 160-). Please, in experimental part change the scheme to elucidate step by step preparation of the new sensor. Please, specify an origin of used materials. Who produced the fiber Bragg grating? It is a commercial product or made in a laboratory? Please, specify an instrument used for braiding. Figure 3 – the snaps are too small.

It is impossible to evaluate a novelty and scientific message of your work after reading the manuscript. Please, add experimental details and rephrase the text.

Author Response

Response to Reviewer 1 Comments

Dear reviewer

We are grateful for the opportunity to resubmit our manuscript. We appreciate you for the thoughtful comments and those help to improve our manuscript. The reviews contained many useful comments and corrections.

As you see below, we have responded in detail to each comment. If there is any insufficient response, please let us know. We supplement it right away.

Best Regards,

Joo Hyeon Lee/song bi Lee

-------------------------------------------------------------------------------------------

--------------------------------------------------------------------------------------------

1.Answers to the correction are described in red text.

3.The text before revision is marked with black text and grey highlights.

4.The location of the revised text is marked in blue text

5.The revised content is marked with yellow highlights on the revised paper and attached.

--------------------------------------------------------------------------------------------

Point 1: Specialist in sensor technology used to read journal Sensors. Therefore, well known common statements as    Fused silica…………..lines 27- , are redundant. Please, eliminate also similar sentences (e.g. lines 160-).

  • Response 1 for Point 1 [Page1, Line 28]

  • The fused silica was modified with a silica optical fiber.
  • Repeated content has been deleted and corrected.

After modification

  • Silica optical fiber has the characteristics of the high purity of more than 99%, excellent heat resistance, thermal stability, chemical stability, excellent light transmittance, and electrical insulation properties. It is resistant to heat and chemical reactions and is useful as an optical material for lasers.[1]

Before modification

  • Fused silica has the characteristics of the high purity of more than 99.99%, excellent heat resistance, thermal stability, chemical stability, excellent light transmittance, and electrical insulation properties. It is resistant to heat and chemical reactions and is useful as an optical material for lasers.[1]

Point 2: Please, in the experimental part change the scheme to elucidate step by step preparation of the new sensor.

  • Response 2 for Point 2. [Page 6~8, Line 224~257]

  • The contents of sensor production were described and corrected step by step.

After modification

  • 3.1. Prototype sample production process

2.3.1.1 Bragg grating sensor process

Helical core structure optical fiber(Fiber Core™) was used for the same braiding yarn angle array. A femtosecond laser was used for precision Bragg grating processing. In the Femto laser processing specification,Femtosecond laser grating-Pulse repetition rate of 1kHz, output 2w, time width 30fs~2ps variable conditions, center wavelength of 785nm, pulse width of 185fs, Femto laser with maximum output of 1W (model-APRI Femto-k1, Advanced Photonics Research Institute, GIST, south Korea) was used to perform the Bragg grating processing of the phase mask treatment. In the Bragg grating processing, scan-speed is 2 mm/s, the line spacing is 2 ?, and the processing depth (Z) is fixed at -60 ?, and the laser power is changed from 0.3 mW to 3 mW. While processing the Bragg grating, the detection of the signal at 10 kHz of the interrogator was detected in real-time, and the peak generation and shift of the wavelength were observed.

Figure 3. Photo (a) is a photorealistic view of processing Bragg gratings in a femtosecond laser precision processing stage using a phase mask. Photo (b) is a real picture of the phase mask (period:2132nm, type: uniform spacing:2nm, size:25×3mm, illumination:808nm) used for femtosecond laser-processing.

2.3.1.2 Bragg grating sensor braiding process

After attaching the epoxy to the fabricated sensor flexible load (cross-section diameter, 1.2?? length 100cm). Arranged so that polyurethane uniformly covers the flexible load The braiding machine (model-KV-BM12) was bonded together with the core yarn at the core yarn position. It was made with 12-strands polyurethane-thread prototype and a 16- strands polyurethane prototype.

Figure 4. Figure 4 (a) is the overall structure of the braiding machine. Figure 4(b) shows the trajectory of the yarn movement of the braiding machine. Eight groups of braiding yarn tracks and one wrap yarn track used for braiding create a combination of spindle movements by repeating the same speed. The core yarn in Figure c9 is surrounded by a tubular braiding structure by a spindle combination of braiding yarn and warp yarn. The sensor is inserted along with the core yarn and processed into a tubular bladed process.

Point 3: Please, specify an origin of used materials. Who produced the fiber Bragg grating? It is a commercial product or made in a laboratory? Please, specify an instrument used for braiding.

  • Response 3 for Point 3 [Page 6Line 226] [Page 7Line 230] [Page 7Line 248]

  • Corrected by indicating the material of the sensor manufacturing, the model name of the device used, and the process and location of the sensor manufacturing source.

After modification

  • Helical core structure optical fiber (Fiber Core™) was used for the same braiding yarn angle array.

femto laser with maximum output of 1W (model-APRI Femto-k1, Advanced Photonics Research Institute, GIST, South Korea)

The braiding machine (model-KV-BM12) was bonded together with the core yarn at the core yarn position.

Point 4: Figure 4 – the snaps are too small.

  • Response 4 for Point 4. [Page9, Line 283~293]
  • a snapshot of the experimental setup been modified.

(A)

(C)

(B)

Reviewer 2 Report

This work is technically sound and promising. However, the representation of the work needs to be improved. Here are some points which need to be highlighted. I think English writing needs to be improved, too.

  1. What are the benefits of using a helical core fiber for FBG fabrication in terms of sensor performance or fabrication processes? Please elaborate on this point for casual readers.  
  2. What are the advantages of braiding an FBG in 3D space? Please emphasize the rationale of proposing this new sensor design. 
  3. What're the things which distinguish this work from previously reported works? 

Miscellaneous

- Page 1, line 17: What is the meaning of "The sensor core structure is helical of 1.54 pitch.". Can you elaborate on this sentence for readers?

- Page 1, line 26: There is an unnecessary period.

- Page 1, line 26: High purity is a material property? And the purity should be 99.999 % instead of 9.999 %?

- Page 1, lines 27 ~ 34: Some references need to be added.

- Page 2, lines 66 ~ 80: Subscripts are not properly applied to some variables.

- Page 2, line 80: Fonts of the symbols are not consistent.

- Page 2, lines 81 ~ 82: Equation 1 and 2 are in different formats. I think it would be better to be consistent in the format of the equations.

- Periods and spaces are not properly used throughout the manuscript.

Author Response

Dear reviewer

We are grateful for the opportunity to resubmit our manuscript. We appreciate you for the thoughtful comments, and those help to improve our manuscript. The reviews contained many useful comments and corrections.

As you see below, we have responded in detail to each comment. If there is any insufficient response, please let us know. We supplement it right away.

Best Regards,

Joo Hyeon Lee/song bi Lee

Round 2

Reviewer 1 Report

The manuscript is still too long with many redundant sentences. In revised version experimental part was improved but the whole manuscript needs reformulation and shortening. You did only minor changes. Common sentences, in some parts whole paragraphs (e.g. lines 73-109) must be erase. These do not belong to scientific paper. They looks like copied from a extbook for undergraduates. You must consult the manuscript with experienced scientist to rephrase the text to be clear the message of the manuscript, the experimental part will not be mixed with results and discussion, and the length of the text will be appropriate of its scientific content.

Author Response

Dear reviewer 1

We are grateful for the opportunity to resubmit our manuscript. We appreciate you for the thoughtful comments, and those help to improve our manuscript. The reviews contained many useful comments and corrections.

As you see below, we have responded to each comment.

Best Regards,

Joo Hyeon Lee/ Song Bi Lee

â–¶ [1] (adjusted)

Parts line 73-109 have been erased.

â–¶[2](adjusted)

  1. The overall composition has been revised, and the results and discussion sections have been. Fixed.

Overall, all text has been revised, and the order of the text has been modified for the flow of the paper.

  1. This paper has been proofread from MDPI.

Reviewer 2 Report

Most of my comments are well addressed. The format of Equation 1 can be matched with that of Equation 3.

Some of the sentences in the paper are still not clear. I strongly recommend the authors to get the paper proofread by someone who is more familiar with English writing.

Author Response

Dear reviewer 2

We are grateful for the opportunity to resubmit our manuscript. We appreciate you for the thoughtful comments and those help to improve our manuscript. The reviews contained many useful comments and corrections.

Best Regards,

Joo Hyeon Lee/ Song Bi Lee

point1.“The format of Equation 1 can be matched with that of Equation 3.”

â–¶ (adjusted)

Remodified to match the format of the formula.

Page2 line[76,82,84]

point2.the paper proofread by someone who is more familiar with English writing.”

â–¶ (adjusted)

This paper has been proofread. from Mdpi.
